# Comet Test in Saliva Leukocytes of Pre-School Children Exposed to Air Pollution in North Italy: The Respira Study

**DOI:** 10.3390/ijerph17093276

**Published:** 2020-05-08

**Authors:** Claudia Zani, Elisabetta Ceretti, Ilaria Zerbini, Gaia Claudia Viviana Viola, Francesco Donato, Umberto Gelatti, Donatella Feretti

**Affiliations:** Department of Medical and Surgical Specialties, Radiological Sciences and Public Health, University of Brescia, 11 Viale Europa, 25123 Brescia, Italy; claudia.zani@unibs.it (C.Z.); elisabetta.ceretti1@unibs.it (E.C.); ilaria.zerbini@unibs.it (I.Z.); gaia.viola@unibs.it (G.C.V.V.); francesco.donato@unibs.it (F.D.); umberto.gelatti@unibs.it (U.G.)

**Keywords:** children, early biological effects, DNA damage, comet assay, mucosa buccal cells, saliva leukocytes, urban air pollution

## Abstract

Air pollution is a well-known problem for human health, especially for children living in highly polluted urban areas. This study aimed to assess the relationship between airborne pollutants concentration and biomarkers of DNA damage in the buccal mucosa cells of pre-school children. DNA damage was investigated with comet test in saliva leukocytes taken from sputum of 3- to 6-year-old children living in Brescia, Northern Italy, collected during two consecutive winter seasons (2012–2013). The daily levels of PM10, PM2.5, NO_2_, CO, SO_2_, benzene and O_3_ in urban air were collected for the whole period. A questionnaire filled in by the children’s parents was used to evaluate indoor and outdoor exposure. DNA damage in saliva leukocytes was evaluated in 152 children and the means of tail intensity and visual score as DNA damage were 6.2 ± 4.3 and 182.1 ± 30.9, respectively. No demographic and indoor or outdoor exposure variable was associated with the two measures of DNA damage. No significant association between air pollution and DNA damage in children’s buccal leukocytes was found. In this study, the comet assay does not appear to be a valuable biomarker to detect DNA damage in children exposed to high levels of air pollutants, such as PM10, PM2.5 and NO_2_.

## 1. Introduction

Air pollution is a global problem: airborne or deposited pollutants can be found worldwide, from highly polluted to remote areas. Particulate matter (PM) has been widely investigated, especially in urban contexts, to determine its potential effects on human health [1,2,3,4,5,6,7]. Epidemiological studies have found a consistent association between exposure to airborne PM and the incidence and mortality for cardiovascular disease and lung cancer [8,9,10,11] and recently also with diabetes and other chronic diseases, possibly through oxidative stress and inflammation [12,13].

For a more complete evaluation of the risk to which people are exposed, it is important to evaluate the global effect of pollutants.

Short-term mutagenicity assays can be used to detect DNA damage resulting from exposure to mutagens. Their use is increasing for biomonitoring people, since they allow the detection of early effects of chronic exposure to toxic agents [14,15], they are valuable for exposure to low doses and mixtures of toxicants, and they require a lower number of subjects and evaluation of objective parameters compared to traditional epidemiological studies [16].

Biomarkers of DNA damage can be investigated in various organs, tissues and body fluids, such as leukocytes or lymphocytes in peripheral blood [17,18,19]. However, cellular suspensions derived from target tissues are usually more appropriate. In particular, exfoliated buccal and nasal cells have been used in biological monitoring of people exposed to airborne pollutants as they are representative of epithelial respiratory tract cells and are easier to collect than those of other respiratory organs [20,21,22,23,24,25,26].

The single cell gel electrophoresis test (SCGE or comet assay) is a mutagenicity test that rapidly detects DNA damage in eukaryotic cells showing a very early, reversible DNA damage [27,28,29,30].

A significant association was found between high levels of urban pollution and DNA damage detected by the comet assay in human lymphocytes [18] and nasal mucosa cells in adults [25].

Children are more vulnerable to the adverse effects of air pollution due to their small body size, fast growth rate and relatively immature organs (lungs in particular), body functions [31], immune system and cell repair mechanisms [32,33]. They have higher levels of physical activity, spend more time outside and have a higher air intake than adults [34]. Moreover, some data suggest that genetic damage, caused by environmental pollutants, viruses or lifestyle factors, occurring early in life can increase the risk of carcinogenesis in adulthood [35].

Although the association between air pollution and respiratory symptoms and diseases has been shown consistently in children as well as adults, few data are available on biomarkers of early effect of air pollution in children [16,36,37,38].

This study is part of the Respira study (the Italian acronym for Rischio ESPosizione Inquinamento aRia Atmosferica) that aimed to assess the relationship between airborne pollutants concentration, urban air genotoxicity and biomarkers of DNA damage in the buccal mucosa cells of pre-school children [20,39].

In this paper, we report the results of the comet test carried out in the saliva leukocytes of 3-to 6-year-old children living in a town with high levels of air pollutants. We also investigated the associations between biological data and the concentrations of some air pollutants and other exposure variables, including indoor pollutants.

## 2. Materials and Methods

The Respira study was described in detail in Ceretti et al., 2014 [20]. This part of the study concerns evaluation of DNA damage in saliva leukocytes of pre-school children living in Brescia, a town with heavy air pollution in Northern Italy. Briefly, we recruited the children attending 6 schools located in different areas of the town, characterized by different levels of traffic and presence of factories. We enrolled children aged 3–6 years, born in Italy and with European parents, without malignant tumours, who had not undergone radiotherapy or chemotherapy in the previous 12 months or X-rays in the previous 3 months. Only Caucasian children were enrolled in the study to limit the variability of response of cytogenetic markers, which may be also influenced by the ethnic group due to genetic polymorphisms and diet [40,41].

DNA damage was investigated in saliva leukocytes taken from sputum: the children rinsed their mouths twice with mineral water and the mouthwashes were collected in tubes containing 25 mL of saline solution (NaCl 0.9%) in order to obtain leukocytes for the comet assay [42].

The biological samples were collected in two consecutive winter seasons during or after a series of days with high levels of fine particulate matter (PM10 and PM2.5). The data about levels of ambient air pollutants were retrieved from the Regional Agency for Environmental Protection (ARPA) database. The daily levels of PM10, PM2.5, NO_2_, CO, SO_2_, benzene and O_3_ from January to March of 2012 and 2013 were collected.

The project was approved by the Ethics Committee of the Local Health Authority of Brescia, Lombardy Region, on 10 August 2011, with the number of the general protocol 0112487.

The children’s parents provided their written informed consent to participate in this study. All the data collected were treated in accordance with the Italian Privacy Law (196/2003).

Children and their parents were interviewed on the characteristics of area of residence (e.g., traffic, factories), and children’s possible exposure to indoor air pollution (stove, fireplace, parental smoking, etc.) using an ad hoc questionnaire.

### 2.1. Comet Assay

Cell suspension in saline solution were centrifuged for 10 min at 1100× *g* (at 4 °C) and the pellet was re-suspended in 1 mL of phosphate-buffered saline (PBS). Leukocyte viability was determined using the trypan blue technique; epithelial buccal cells present in the sputum samples were excluded. The comet assay was performed on buccal leukocytes, according to the method of Singh et al., 1988 [43] with minor modifications. PBS cell suspension was centrifuged for 4 min at 8700× *g*, and the pellet was re-suspended in 200 µL of LMA (low melting point agarose, 0.7%) and layered onto pre-treated (NMA, normal melting point agarose, 1%) slides. After overnight lysis (4 °C) in an alkaline buffer (pH 10) of cellular and nuclear membranes, the slides were placed in a horizontal electrophoresis box, allowed to unwind for 20 min in an electrophoretic alkaline buffer (pH > 13) and then subjected to electrophoresis (4 °C) for 20 min by applying an electric field of 0.8 V/cm and adjusting the current to 300 mA. Lastly, the microgels were neutralized and fixed with absolute ethanol (−20 °C). To evaluate DNA damage, the slides were stained with ethidium bromide (10 µg/mL) and examined using a fluorescence microscope equipped with a high-sensitivity CCD (charge-coupled device) camera connected to a computerized image analysis system (Komet 5, Kinetic Imaging Ltd.). Computerized imaging was performed on coded slides using dedicated software which estimates damage parameters (e.g., tail length and tail intensity) by comet profile. DNA migration was evaluated also by visual score, a parameter derived from visual classification of the comets into four damage classes. Two hundred cells were analyzed for each subject (100 cells/slide, 2 slides per subject).

### 2.2. Statistical Analysis

The concentrations of air pollutants and DNA damage biomarkers are reported as means, standard deviations (SD), medians and percentages. We investigated the association between the visual score and tail intensity of the comet assay and the mean air pollutants levels one day, two days and one week before the biological sampling. Regression analysis was also performed with DNA damage as the dependent variable and the levels of air pollutants as predictors, adjusting for confounding factors (age, parents’ smoking and education, traffic near home, and others). Two-tailed statistical tests were performed with 0.05 *p*-value as the threshold for rejecting the null hypothesis. All the analyses were performed using the Stata TM 12.0 statistical package (Stata Statistical Software Release 12.0, 2012; Stata Corporation, College Station, TX, USA).

## 3. Results

A total of 222 children were enrolled, for 152 of which biological samples were adequate for the comet assay. None of the participants had asthma or other respiratory disorders investigated with the questionnaire.

The means of tail intensity and visual score were 6.2 ± 4.3 and 182.1 ± 30.9, respectively. No difference was observed in the comet test among children attending different schools or residing in different parts of the town.

In Table 1 the results of the comet test expressed as visual score and tail moment, according to demographic characteristics and children’s exposure variables are reported. No demographic and exposure variable was associated with the two measures of DNA damage in saliva leukocytes.

The results of linear regression analysis of the DNA damage, expressed as visual score and tail intensity, on the concentration of air pollutants (PM10, PM2.5, NO_2_, SO_2_, benzene and O_3_) at 1, 2 and 7 days preceding biological sampling showed that the ozone level measured two days, but not one or 7 days, before the sampling was associated with an increase in DNA damage, which was statistically significant with the Wald test on the coefficient (*p* = 0.05). These results, considering the visual score data, are shown in Table 2.

Figure 1 shows the scatterplots and the regression linear of the visual score by ozone concentration at each time point.

By contrast, the concentration of SO_2_ measured the day before the sampling showed a negative association with DNA damage (*p* = 0.04). No pattern was evident for the other air-quality parameters.

## 4. Discussion

This study focused on a biomarker of DNA damage in children because some data suggest that genetic damage occurring early in life may influence the risk of carcinogenesis and other chronic diseases in adulthood more than later damage. The use of a direct measure of biological effect, i.e., DNA breakages, which can be detected a long time before clinical disease develops can give us useful information about the substances to which children are exposed. This study did not show any association between DNA damage in buccal leukocytes of pre-school children and air pollutant levels at 1, 2 and 7 days before the sampling, apart from ozone: the mean ozone levels in the two days before the sampling was weakly associated with DNA damage, next to the limit of statistical significance, whereas the mean concentration of this pollutant in the day and in the 7 days preceding the sampling was not.

The comet test detects transient lesions produced directly by the mutagenic agent or indirectly by cellular repair mechanisms, and therefore is suitable to assess the effects of recent exposures. For this reason, we considered the concentration of the pollutants in the days immediately preceding the biological sampling in children.

Also the demographic variables and those concerning indoor pollution (passive smoking, parents’ level of education, traffic next to children’s school and home) were not significantly associated with comet visual score and tail intensity parameters. On the other hand, our indirect assessment of other children’s exposures through a parental interview may have determined an imprecise estimate of the relationship with children’s DNA damage.

Brescia is a town in North Italy with a high level of ambient air pollutants in winter, and during the period of biological sampling the concentrations of PM10 and PM2.5 were almost always over the European Union (EU) limit values for daily means. On the contrary, the concentration of other pollutants were always below the EU limits (data reported in Ceretti et al., 2014 [20]). In these children, a very high level of micronuclei (MN) frequency in exfoliated buccal cells was observed [20] which was associated with high levels of fine particulate matter. Therefore, these two biomarkers might be suitable to assess the effects of different pollutants: the comet test might reveal an oxidative damage possibly linked to the ozone concentration, according to others [17,25] while the micronuclei test might reveal more stable damage induced by substances on fine particles, such as polycyclic aromatic hydrocarbons [38]. It is of note that we found no correlation between micronuclei frequency and DNA damage detected with the comet test in these children, suggesting that these biomarkers can detect independent events, as shown by other biomonitoring studies [44].

The concentration of ozone during the study period was not particularly high as this is a typically summer pollutant. Unfortunately, the study did not provide a collection of biological samples in children in periods with high concentration of oxidizing compounds such as ozone.

No difference was observed in DNA damage measured with comet test in salivary leukocytes of children attending schools located in various areas of the town, suggesting that residence or school area did not affect DNA damage, as observed for our study on micronuclei frequency [20]. Accordingly, we found a uniform distribution of pollutants in the urban air our town as well as mutagenicity of the particulate matter collected in different sites of the town in previous studies [20,39,45].

Child exposure to air pollution is a matter of concern because of the well-known health effects of airborne pollutants and the relevance of early damage in childhood on the risk of developing chronic degenerative diseases in adulthood. Several studies have investigated the effects of outdoor and indoor airborne pollutants on children, using indirect measures of subject exposure, such as subjects report of vehicle traffic in the next-by streets, or direct measures of air pollutants but referred to a larger area, such as a whole town, and most of them have used the presence of symptoms or respiratory diseases as the effect measure, often based on self-referral. The Respira study, instead, used biomarkers of early biological effect, such as micronuclei formation [20] and DNA damage detected with the comet assay in children’s buccal cells.

Various studies on adults exposed to high levels of air pollution using various biomarkers of biological effects in peripheral leukocytes found an association between air pollution exposure and DNA damage [14,16]. Although children are usually considered to be more susceptible to DNA damaging chemical compounds than adults [16], there is very little evidence that these or other early effect biomarkers are related to urban air pollution exposure in children as well in adults at present [44,46,47,48,49,50,51,52,53,54].

The results of this research are not fully comparable with other studies because we included very young children and used different comet parameters (e.g., tail intensity, tail length, olive tail moment). Moreover, little data are available in the buccal cells of very young children. Nevertheless, the mean value for tail intensity found in our children was 6.2 ± 4.3, that is of the same order of magnitude as that observed in other studies in peripheral lymphocytes of children and adolescents living in low- and high-polluted Brazilian urban areas [44,54]. The correlation between chromosomal damage in peripheral lymphocytes and that in buccal cells [55] makes salivary leukocytes a valid alternative to blood lymphocytes.

We observed a weak linear relationship between ozone concentration and DNA damage through the comet test, which is difficult to interpret because of the low levels of this pollutant in winter in our area. However, the lack of sampling collection in summer, when ozone levels are high, does not allow us to draw a definite conclusion on this point. The absence of data for summer, when most air pollutants except ozone are lower than in winter, and the lack of a control group of subjects living in a less polluted area are a limitation of this study. Another limitation of the study is that biological sampling was carried out only once for each child, in line with other studies investigating the association between air pollution exposure and biological damage in the general population [25,26,44,46,54]. The sampling repetition at different times could provide useful information on the variability of the relationship between cellular damage and air pollution. Nevertheless, the strength of this study is that DNA damage in buccal leukocytes was analysed in 152 children living in the town and exposed to high levels of air pollutants in winter which is a not negligible number when compared to present research with this biomarker [44,46,48,50,52].

New and interesting approaches that use exhaled breath, a non-invasive technique now used above all for diagnostic purposes, could also have applications in vitro tests and be combined with other biomarkers to evaluate the risks in children of future respiratory diseases [56,57,58,59]. The non-invasiveness is an important aspect in preventive studies involving children. Leukocytes from saliva as well as epithelial cells of the oral mucosa, representative of epithelial respiratory tract cells, were easy to collect, and especially in the pediatric population, a collection method that is more acceptable to children and their parents which favored participation in the study. Furthermore, the correlation between chromosomal damage in peripheral lymphocytes and that in buccal cells makes salivary leukocytes a valid alternative to blood lymphocytes.

## 5. Conclusions

In this study no significant association between air pollution and DNA damage in children’s saliva leukocytes was found. The comet test does not appear to be a valuable biomarker to detect DNA damage in children exposed to high levels of air pollutants, such as PM10, PM2.5 and NO_2_.

## Figures and Tables

**Figure 1 ijerph-17-03276-f001:**
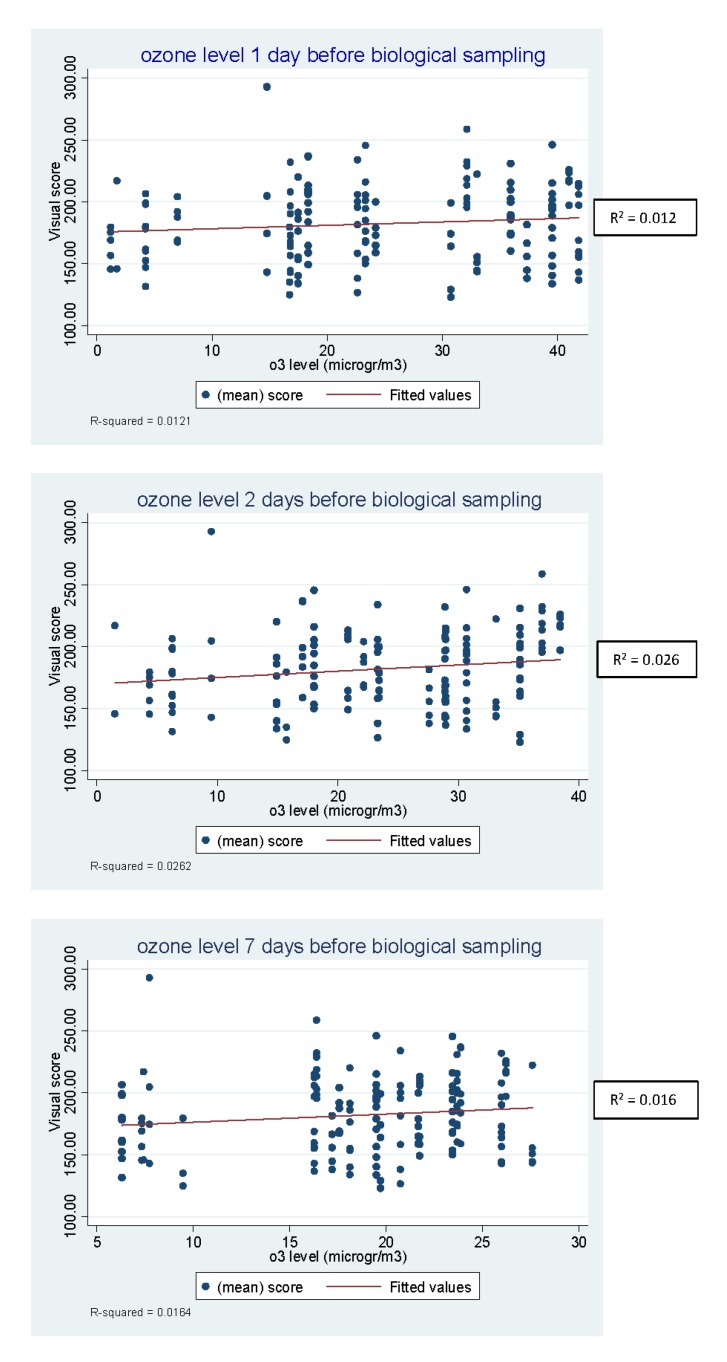
Scatterplots and the linear regression of the visual score by ozone concentration at 1, 2 and 7 days before biological sampling.

**Table 1 ijerph-17-03276-t001:** Comet test results expressed as visual score and tail intensity according to demographic and indoor and outdoor environmental exposure variables.

Demographic and Indoor and Outdoor Environmental Exposure Variables	N (%)	Visual Score		Tail Intensity %	
		Mean ± SD	Median	Mean ± SD	Median
Sex					
M	85 (55.9)	180.4 ± 30.8	179.0	6.1 ± 4.6	5.8
F	67 (44.1)	184.2 ± 31.2	187.5	6.4 ± 3.9	7.0
Children’s age					
3 years	26 (17.1)	190.8 ± 30.1	196.2	7.1 ± 3.8	7.5
4 years	52 (34.2)	177.7 ± 33.1	171.8	5.5 ± 4.6	4.4
5–6 years	74 (48.7)	182.1 ± 29.4	184.3	6.3 ± 4.2	6.8
Parents’ education (at least one parent)					
Primary school or less	16 (10.5)	175.3 ± 28.9	175.4	5.3 ± 3.7	4.5
Secondary school	47 (30.9)	184.8 ± 29.1	187.3	6.8 ± 4.2	7.4
College or university	89 (58.6)	181.9 ± 32.3	179.0	6.0 ± 4.4	5.4
Home characteristics					
*Traffic in the area*					
Heavy	86 (56.6)	181.4 ± 29.3	182.5	6.0 ± 4.0	6.5
Moderate	49 (32.2)	185.7 ± 33.4	180.0	6.8 ± 4.8	7.2
Very light	17 (11.2)	175.2 ± 32.0	168.4	5.4 ± 4.4	4.1
*Truck traffic in the area*					
Heavy	27 (18.0)	171.6 ± 26.5	164.0	4.7 ± 3.5	3.3
Moderate	59 (39.3)	180.6 ± 27.6	181.5	5.9 ± 4.1	6.3
Very light	64 (42.7)	188.9 ± 34.2	190.9	7.2 ± 4.6	7.7
*Indoor exposure*					
Gas stove in home	7 (4.6)	172.9 ± 18.9	169.0	4.6 ± 2.8	4.6
Fireplace in home	34 (22.4)	177.6 ± 31.8	179.5	5.6 ± 4.0	5.2
Presence of smokers in home	24 (15.8)	179.2 ± 31.4	173.7	5.3 ± 4.5	3.7
School characteristics					
*Traffic in the area*					
Heavy	87 (58.0)	178.0 ± 29.1	175.1	5.6 ± 3.9	5.3
Moderate	54 (36.0)	187.0 ± 33.0	185.3	7.0 ± 4.8	7.2
Very light	9 (6.0)	188.4 ± 35.4	198.0	6.7 ± 4.4	8.7
*Truck traffic in the area*					
Heavy	26 (17.4)	175.4 ± 28.7	169.7	4.9 ± 3.7	4.1
Moderate	64 (43.0)	181.1 ± 28.7	183.7	6.1 ± 4.0	6.5
Very light	59 (39.6)	186.4 ± 34.3	185.5	6.8 ± 4.7	7.4
Child’s habits					
*Plays outdoors*					
Less than 1 h	66 (44.0)	181.0 ± 31.3	179.2	6.2 ± 4.4	6.1
More than 1 h but less than 3 h	57 (38.0)	182.9 ± 28.9	187.3	6.3 ± 4.3	6.6
3 h or more	27 (18.0)	184.4 ± 35.4	185.1	6.4 ± 4.2	7.3
*Staying in the kitchen while meals are cooked*					
Never	19 (12.7)	182.1 ± 30.1	181.5	5.8 ± 4.1	5.3
Sometimes	97 (64.6)	180.1 ± 29.5	180.0	5.9 ± 4.1	6.0
Often/always	34 (22.7)	186.7 ± 35.9	179.2	7.0 ± 5.1	6.5
Parents’ smoking habits					
Neither parent smokers	97 (64.2)	183.9 ± 32.5	186.0	6.5 ± 4.4	7.0
Mother smoked during pregnancy	32 (21.0)	185.3 ± 23.7	187.4	6.3 ± 3.4	6.7
Mother smoker	27 (17.8)	178.0 ± 23.3	179.5	5.2 ± 3.4	4.8
Father smoker	43 (28.3)	177.7 ± 28.5	173.2	5.5 ± 4.1	4.6
Both parents smokers	38 (25.2)	178.6 ± 22.1	179.2	5.4 ± 3.4	4.7

**Table 2 ijerph-17-03276-t002:** Coefficients of linear regressions of the DNA damage expressed as visual score on the concentration of air pollution at 1, 2 days and 1 week preceding biological sampling computed for units of increase for each pollutant.

Pollutant Levels in Days before Sampling	Coefficient	95% Confidence Interval	*p*-Value
PM10 1 day	0.035	−0.112; 0.183	0.63
PM10 2 days	0.008	−0.179; 0.195	0.93
PM10 7 days	0.085	−0.171; 0.341	0.51
PM2.5 1 day	−0.036	−0.211; 0.138	0.67
PM2.5 2 days	−0.053	−0.262; 0.155	0.61
PM2.5 7 days	0.176	−0.166; 0.518	0.31
Benzene 1 day	−3.96	−8.947; 1.015	0.11
Benzene 2 days	−3.89	−9.018; 1.223	0.13
Benzene 7 days	−1.01	−7.336; 5.310	0.72
NO_2_ 1 day	−0.096	−0.449; 0.257	0.59
NO_2_ 2 days	−0.171	−0.639; 0.297	0.47
NO_2_ 7 days	0.246	−0.304; 0.797	0.37
SO_2_ 1 day	−1.57	−3.095; −0.045	0.04
SO_2_ 2 days	−0.53	−2.487; 1.418	0.58
SO_2_ 7 days	0.61	−1.460; 2.684	0.58
O_3_ 1 day	0.27	−0.134; 0.690	0.18
O_3_ 2 days	0.51	0.001; 1.019	0.05
O_3_ 7 days	0.63	−0.215; 1.491	0.14

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
