# Peer review of "Comet Test in Saliva Leukocytes of Pre-School Children Exposed to Air Pollution in North Italy: The Respira Study"

_ijerph, 2020, doi:10.3390/ijerph17093276_

Round 1

Reviewer 1 Report

The authors reported their experience on the relationship between airborne pollutants concentration and DNA damage in the buccal mucosa cells of pre-school children.

They retrospectively evaluated 152 children who lived in Brescia, Northern Italy.

They couldn't find any significant association between air pollution and DNA damage.

Overall, the paper is interesting. 

Some concerns should be addressed by the authors.

- The analyses were carried out in the saliva of children aged 3-6. Since the authors are interested in air pollution, I was wondering why they did not evaluate the volatile organic compounds or the exhaled breath condensate in this patients' cohort. In my opinion it would have been more appropriate than the saliva examination. Could the authors comment on that?

With this regard, these two papers might be considered and possibly cited by the authors. They might be also considered these techniques in the introduction and discussion of the manuscript.

  • Bianchi F, Riboni N, Carbognani P, Gnetti L, Dalcanale E, Ampollini L, Careri M. J Pharm Biomed Anal. 2017 Nov 30;146:329-333.  Solid-phase microextraction coupled to gas chromatography-mass spectrometry followed by multivariate data analysis for the identification of volatile organic compounds as possible biomarkers in lung cancer tissues.

Corradi M, Poli D, Banda I, Bonini S, Mozzoni P, Pinelli S, Alinovi R, Andreoli R, Ampollini L, Casalini A, Carbognani P, Goldoni M, Mutti A. Exhaled breath analysis in suspected cases of non-small-cell lung cancer: a cross-sectional study.J Breath Res. 2015 Jan 29;9(2):027101. doi: 10.1088/1752-7155/9/2/027101.

The authors did not use "pure saliva" but they took samples of mouthwashes with mineral water. The authors should comment on that and explain the reasons of such a choice. Did the authors know the effect (if any) of mineral water on fine particulate matter? What about distilled water?

The authors should clearly stated the number of samples per patient they collected during the study period. The authors reported that "the children rinsed their mouths twice with mineral water and the mouthwashes were collected...". Did they perform this procedure only one single time per patient? If so, I think it should have been repeated several times. Could the authors comment on that?

  • Why did the authors choose january-march and not november or december? Why didn't they take samples once a month?
  • I was wondering why the authors decided to publish these data after so many years. Could the authors comment on that?
  • How many of these patients developed an allergic asthma? Did the authors look at that? In my opinion that might be a very interesting point for people studying air pollution.
  • Did the authors follow-up those patients? They will be 10-13 years old at the moment. So it might be very attractive to know their state of health.
  • The authors should report the approval of the study by the local Ethical Committee.

Reviewer 2 Report

This study examined DNA damage measured by comet assay in buccal leukocytes from 152 children during high pollution winter months over two consecutive years. No strong correlations of measured or inferred pollution correlated with DNA damage measured by the assay. I believe one area of importance of the study is that it shows, in a large cohort, that this approach may be insensitive to DNA damage caused by pollution – this needs to be published so other researchers can evaluate the claim for themselves before launching into similarly large efforts, and potentially adopting more sensitive methods instead. Also, the study (less confidently) raises the possibility that air pollution is not associated with DNA damage in children of this age and realm of exposure, specifically in buccal leukocytes. That being said, there was a noticeable trend with ozone pollution that approached significance to varying degrees at all time points tested. This shouldn’t be completely ignored even if the p-value threshold was not met. I also believe that the authors should further examine their study limitations before this is published.

I have the following specific comments and concerns I hope to see addressed:

-Line 39: I believe the authors mean “toxicants” instead of “toxics”

-Line 73: What is meant by “with European parents”? European citizenship? This is vague and needs to be re-stated to clarify what the authors meant.

-Authors should clearly justify the choice to study buccal leukocytes from a toxicological perspective, as opposed to other cells.

-What was the basis of the age range selected? Is it possible that cumulative damage could be more sensitively detected in an older age range?

-Please change instances of “buccal cells” to “buccal leukocytes” to clarify that epithelial cells are not being studied here. Even though this was specified in the text, it is still a bit confusing.

-The ozone data suggest a modest trend could be possible at all time points. (Technically, none were significant – p=0.05 is not a significant result!) I suggest the authors provide scatterplots for each time point so readers can visualize individual data points for themselves, or at least include this for reviewers/editors of this journal.

-Was the regression analysis parametric? If so, were the data normally distributed (how was that tested)? If not, consider Spearman’s correlation instead.

-Authors should discuss whether a number of alternative means of assessing DNA damage, in cells and biological fluids, would have been more sensitive.

-Was the same subject ever studied twice in the two consecutive years or was each subject of the 152 unique?

-Was there any presence of lung disease (e.g. asthma) in a subset of the children, and any association of pulmonary health and pollution levels or comet assay results?

Round 2

Reviewer 1 Report

The authors should be congratulated for their review.